# Timing of Adiposity Rebound and Determinants of Early Adiposity Rebound in Korean Infants and Children Based on Data from the National Health Insurance Service

**DOI:** 10.3390/nu14050929

**Published:** 2022-02-22

**Authors:** Eun Kyoung Goh, Oh Yoen Kim, So Ra Yoon, Hyo Jeong Jeon

**Affiliations:** 1Human Life Research Center, Dong-A University, Busan 49315, Korea; ekgoh72@gmail.com (E.K.G.); oykim@dau.ac.kr (O.Y.K.); 2Department of Food Science and Nutrition, Dong-A University, Busan 49315, Korea; 3Department of Health Sciences, Dong-A University, Saha-gu, Busan 49315, Korea; 4Institute of Health Insurance and Clinical Research, National Health Insurance Service Ilsan Hospital, 100 Ilsan-ro, Ilsandong-gu, Goyang-si 10444, Korea; ysr09@nhimc.or.kr; 5Department of Child Studies, Dong-A University, Busan 49315, Korea

**Keywords:** adiposity rebound, obesity, determinant, nationwide

## Abstract

Adiposity rebound (AR) is defined as the second rise in the body mass index (BMI) usually occurring in early childhood. This study aimed to investigate the timing of AR and the factors determining early AR (EAR) by tracking BMI patterns using large-scale longitudinal nationwide data (*n* = 142,668; 73,389 boys and 69,279 girls) over seven time periods (4–6, 9–12, 18–24, 30–36, 42–48, 54–60, and 66–71 months). The average BMI rebound indicating AR was found before the age of 5 years (6th time period, 54–60 months). Interestingly, children experiencing BMI rebound during the 4th to 6th time periods showed a small increase in the proportion of underweight in the 2nd time period, but a dramatically higher proportion of underweight during the corresponding time period, similar to M-shaped patterns. In contrast, overweight or obese children in the above group showed opposite W-shaped patterns. The parameters predicting the risk of EAR are diverse for each time period by sex. Adequate breastfeeding before the age of 1 year, adequate diet, and reduced sugar-sweetened beverage consumption after the age of 1 year were important for reducing EAR. This study presents for the first time, the timing of AR and the major determinants of EAR among Korean infants and children based on large-scale nationwide data.

## 1. Introduction

The proportion of obesity among Korean children and adolescents has been continuously increasing by 0.5% and more every year [1]. It was reported that the proportion of obesity in boys and girls at the age of 6 years was 8.1% and 8.9%, respectively, which rapidly increased up to 29.2% and 18.8% at the age of 18 years, respectively [1]. Childhood obesity is known to be the main cause of metabolic syndrome and chronic diseases such as hypertension, dyslipidemia, insulin resistance, type 2 diabetes (T2DM), and fatty liver disease [2].

Several large-scale cohort studies have demonstrated that the timing of adiposity rebound (AR) is an important indicator of the development of obesity after school age [3,4,5,6,7,8,9]. According to a report by Rolland-Cachera et al. [4], AR is defined as the second rise in body mass index (BMI) usually occurring in early childhood. For example, the average age of AR was 5.5 years in the US cohort data [9], and 6.6 years for boys and 6.0 years for girls in the New Zealand cohort data [10]. On the other hand, the age showing rebound of average BMI was 4.3 years for boys and 4.7 years for girls in Korea based on the 2005 Korean National Children and Adolescents Growth Survey, which were a little earlier than those observed in the US and New Zealand cohort data, but the average age of AR was 5.5 years in both boys and girls at the 50th percentile based on the 2017 Korean National Growth Charts for children and adolescents [10,11]. Cohort studies have reported that AR at an early age is a risk factor for later obesity [4,6,7,8,9]. AR occurring before 4 years increases BMI in adolescence [6,7,8], and the increased BMI is maintained in early adulthood [7], leading to obesity in adulthood [9]. AR before the age of 4 years was also reported to increase the risk of metabolic syndrome including insulin resistance, increased cholesterol, and elevated blood pressure in the early teenage years [3], and shows a higher proportion of atherogenic small dense LDL at the age of approximately 12 years [12]. However, AR occurring before the age of 5.5 years was also reported to increase the risk of obesity in early adulthood [3], and earlier AR that occurs before the age of 5.4 years might lead to obesity and T2DM in young adult males [10]. Another study also reported that AR that occurs before the age of 5 years is associated with T2DM in both men and women [13]. Therefore, it is necessary to analyze the patterns and factors affecting early AR to prevent obesity-related diseases and metabolic syndromes.

Recently, obesity rates have increased among East Asian children, particularly in China and Korea [1,14]. Studies have also compared the timing of AR between East Asian and Western children [14,15,16]. The timing of AR among Chinese children was earlier than that of American and British children, which was thought to be associated with early changes in eating habits and lifestyles according to the development of the Chinese economy [15]. As the percentile of BMI is higher, the timing of AR becomes earlier. Children who had earlier onset of AR have experienced secondary sexual characteristics faster than those who had normal onset of AR, with rapid physical growth and maturation [15], which is similar to the characteristics observed in obese children [16]. On the other hand, the process of rapid compensation for deficiencies in nutrition or growth retardation during the early stage of development is thought to be one of the causes of early AR, in addition to high calorie consumption, fast physical growth or obesity during early childhood [4,17,18]. For example, children born with lower body weight [16] and smaller head circumference [18] which are representative body indices affected by the fetal environment are sensitive to physical recovery, thereby having AR at an earlier stage [17,18]. Nutritional imbalances after birth can also result in compensatory body growth [4]. For example, among children with similar BMIs, those experiencing nutritional imbalance such as high protein and low-fat intake during the infant period, have increased susceptibility to dietary fat. This easily leads to excess energy intake and accumulation of fats in the body, which may contribute to early AR [4]. Consequently, body mechanisms developed by overcoming the risk of morbidity during the fetal period or early life can easily lead to overnutrition and cause health problems (i.e., impaired glucose tolerance, coronary artery disease, atherosclerotic lipid profile, blood coagulation dysfunction, obesity, stress-sensitive reaction etc.) in adulthood or in an environment supplied with sufficient nutrients [19,20]. Therefore, high calorie intake or early obesity may be the result of early AR. To examine the nutritional factors affecting the AR period, non-nutritional factors such as birth weight, preterm birth, and household income need to be controlled [17,21,22]. However, previous studies have not provided exact information on the characteristics of the groups with similar onset of AR because the groups were tracked based on the BMI percentile group [15,16]. Furthermore, most studies comparing groups with different AR onset were small-sized cohorts that tracked the effects in the situation after AR had happened [5,6,7,8,9,12,13]. In addition, a study analyzing the factors determining the onset of AR showed that the baseline BMI of children with early AR account for 50–68% of overweight or obesity [23], so it is difficult to compare the timing of early AR by tracking the groups with different onset of AR when considering the normality of BMI values.

Therefore, this study aimed to determine the nutritional factors affecting early AR onset by tracking the pattern of BMI among groups with different timing of AR by controlling birth weight, preterm birth, and household income using large-scale longitudinal nationwide data.

## 2. Materials and Methods

### 2.1. Study Population and Data Collection

This study was approved by the Institutional Review Board of Dong-A University (2-104709-AB-N-01-201712-HR-054-02) and passed the review of the National Health Insurance Service (NHIS). The data analysis was performed from August 2018 to 22 March 2019. This study used data from the screening program for infants and children (SPIC) provided by the NHIS which tracked infant and children from 4 months to 71 months for seven-time periods (4–6 months, 9–12 months, 18–24 months, 30–36 months, 42–48 months, 54–60 months, and 66–71 months) without missing data of body weight and height nor non-errors in the age of measurement: specifically, 30,776 people born in 2008, 47,172 people born in 2009, and 64,723 people born in 2010 were combined (total 142,671) after checking the average AR onset by each of birth year, which was 54–60 months, indicating no difference in the period of rebound of average BMI. Finally, data from 142,668 people (73,389 men and 69,279 women) were used for the analysis, excluding the extreme outliers of BMI ≤ 4 or ≥50 kg/m^2^.

### 2.2. Parameters

Data collected during the seven screening time periods for screening included body weight, height, nutritional measurements and other environmental parameters. Anthropomentric data were collected from the hospital where the physical measurements of the children for SPIC were performed by trained medical staff, and nutritional data were recorded by the parents of the child using a questionnaire designed by the NHIS. Briefly, the input variables related to nutrition in the questionnaires for each time period were as follows: type of feeding milk (i.e., breast milk, formula milk, mixed milk including breast milk and formula milk, and special formula milk) and start of weaning food before or after the age of 4 months during the 1st time period; number of weaning food per day, timing of the 2nd start of weaning food and type of milk (i.e., breast milk, goat’s milk, soy milk, cow’s milk, and formula milk), and any of ingredients (i.e., grains, vegetables, eggs, fish, meats) omitted in weaning food at the 2nd time period; regular eating at the given place, daily intake of sugar-sweetened beverages, use of salt in food, and vitamin supplementation at the 3rd time period; appetite, number of meals per day, number of snacks (including fruits) per day, daily intake of cow’s milk, daily intake of sugar-sweetened beverage, parents’ concerning about child’s eating habits at the 4th time period; appetite, number of meals per day, daily intake of cow’s milk, daily intake of sugar-sweetened beverage, and picky eating at the 5th time period. The AR onset was confirmed by the average BMI of total population (*n* = 142,668), which rebounded at the 6th time period (54–60 months). Therefore, early AR was defined as the rebound of BMI before the 6th time period. In addition, the average change in BMI among the total population was tracked from the 1st time period to the 7th time period. Based on the average value of BMI at the 7th time period, the BMI changes above or below the average value were also tracked. Changes in the average BMI values among children at each time period were tracked from the 1st time period to the 7th time period. After all BMI data during the seven time periods were checked, the study population was classified into seven AR groups according to the specific time period when the BMI level became minimum and then increased again. The group that showed a continuous decrease in BMI without rebound by the 7th time period was also included as the non-AR group in the seven AR groups. 

Determination of the BMI percentile by age and sex was based on the Child Growth Standards of the World Health Organization (WHO). Specifically, the population was classified as severely underweight (BMI-for-age less than 5th percentile), underweight (5th to less than 15th percentile), healthy weight (15th to less than 85th percentile), overweight (85th to less than 95th percentile), and obese (95th to 100th percentile) based on the age at the time of measurement.

### 2.3. Statistical Analysis 

Statistical analysis was performed using R 3.4. The regression model explored the factors affecting the risk of early AR by comparing the rebound group in each time period with the non-rebound group at 6th time period. Since most regression models have highly predictive values (beta) for sex, it is assumed that the influences of factors explaining the regression model for early AR are diverse according to sex. Therefore, the regression models were calculated separately for males and females. The variables used for adjustment were birth weight, preterm birth, and household income. Variables showing statistical significance or trends in the regression model for each period are presented in the tables. The results were presented as mean ± standard deviation for descriptive statistics, *n*, or % for proportions. A logit model analysis was performed to identify the factors affecting early AR and calculate the odds ratio. 

## 3. Results

### 3.1. Basic Information of Study Population 

Table 1 shows the basic information of the study population (*n* = 142,668). The BMIs of the population born in 2008, 2009, and 2010 were tracked from the 1st time period (4–6 months) to the 7th time period (66–71 months). Among the study population, 51.44% were men. The average birth weight among boys (3.25 ± 0.47 kg) was slightly heavier than that among the girls (3.15 ± 0.45 kg). The household income level and the proportion of preterm birth are similar between boys and girls (11.81 ± 4.46 vs.11.82 ± 4.42; 3.61% vs. 3.72%, respectively). 

As the average BMI rebound of the whole study population was observed at the 6th time period, nutrition-related parameters measured from the 1st time period to the 5th time period were used as input variables for the prediction of early AR. During the 1st time period, proportion of breast milk intake was highest (boys: 44.33%, girls: 48.42%), second highest is formula milk (boys: 33.82%, girls: 32.01%), and then mixed milk (boys: 21.42%, girls: 19.21%) in that order. The proportion of starting weaning food was significantly higher after age of 4 months (boys: 93.15%, girls: 93.45%) than before 4 months. During the 2nd time period, the number of weaning food per day was the highest at three times (boys: 70.62%, girls: 68.46%), and the proportion of feeding milk types was the highest in formula milk (boys: 49.88%, girls: 48.10%) and second highest in breast milk (boys: 38.81%, girls: 40.77%). During the 3rd time period, most of the study population consumed 200 mL or less amount of sugar-sweetened beverages per day (boys: 93.40%, girls: 94.20%), and salt used in the food (boys: 83.90%, girls: 83.30%). On the other hand, the proportion of those taking vitamin supplements was relatively small (boys: 39.36%, girls: 37.85%). During the 4th time period, most children showed normal or good appetite (boys: 94.45%, girls: 94.76%), and consumed meals three times per day (boys: 89.14%, girls: 87.89%). About two-thirds of the population consumed snacks two times per day (boys: 67.82%, girls: 67.48%), and more than a half of the population were drinking 200–500 mL of cow’s milk per day (boys: 61.10%, girls: 59.34%). On the other hand, majority of the population consumed 200 mL or less amount of sugar-sweetened beverages per day (boys: 93.20%, girls: 94.53%). In addition, the proportion of parents concerned about their child’s eating habits was close to 50% (boys: 48.15%, girls: 47.20%). At the 5th time period, most children consumed meals three time per day (boys: 91.34%, girls: 90.04%), and more than a half of the populations were drinking 200–500 mL of cow’s milk per day (boys: 60.44%, girls: 56.90%). On the other hand, majority of the population consumed 200 mL or less amount of sugar-sweetened beverages per day (boys: 93.32%, girls: 94.89%).

### 3.2. Changes of Body Mass Index for Seven-Time Periods 

Figure 1 presents the changes in the average BMI among boys and girls for the seven time periods. It also shows the lowest BMI at the 6th time period in both gender groups (boys: 16.08 ± 1.54, girls: 15.88 ± 1.48). Compared to the average BMI at the 7th time period (boys: 16.24 ± 1.88, girls: 15.96 ± 1.73), those who had higher BMI levels at the same time period (boys: 17.78 ± 1.74, girls: 17.59 ± 1.49) maintained overall higher BMI levels at each time period even from the 1st time period. On the other hand, those who had lower BMI levels (boys: 15.01 ± 0.72, girls: 14.90 ± 0.79) than the average BMI at the 7th time period showed lower BMI levels at each time period, even from the 1st time period. Detailed information on changes in BMI during the seven time periods is presented in Appendix A. 

### 3.3. Distribution of BMI Percentile for the Seven Time Periods According to Each Time Period and AR Onset

As shown above, the average BMI rebound was observed in the 6th time period in the study population. Therefore, the BMI rebound that occurred during the 1st to the 5th time period (before 4 years of age) was classified as early AR (EAR) in this study. The proportion of EARs was 56.34% among the girls and 52.40% among the boys (Appendix A). Appendix A present detailed information on changes in BMI levels for the seven time periods according to each time period of AR by sex.

Figure 2a,b present the distribution of BMI percentile for the seven time periods according to each time period and AR onset by sex (a: boys, b: girls). Among the boys, 60.99% of those who had experienced AR in the 1st time period showed BMI level below the 15th percentile based on the WHO child growth standards which indicates underweight. Specifically, 35.56% of the boys showing BMI levels below the 5th percentile indicating severe underweight in this period belonged to the group experiencing AR in the 1st time period (Appendix A). Interestingly, 57.96% of boys with AR in the 1st time period became overweight or obese indicating BMI levels above the 85th percentile in the 7th time period. Among the girls, 51.0% of those who had experienced AR in the 1st time period had BMI levels below the 15th percentile, and 47.06% of the girls with BMI levels below the 5th percentile indicating severe underweight in this period belonged to experiencing AR in the 1st time period (Appendix A). In addition, more than half of them were underweight in the 1st time period, but 47.82% of them were found to be overweight or obese in the 7th time period. The boys experiencing AR in the 2nd time period (51.32%) were underweight in this time period, but 58.96% of this group became overweight or obese in the 7th time period. Among girls, 44.26% of those with AR in the 2nd time period were underweight in this time period, but 45.44% of this group became overweight or obese in the 7th time period. During the 3rd time period, proportions of underweight among the children experiencing AR in this time period (boys: 25.83%, girls: 19.91%) were relatively higher than those experiencing AR in other time period, but the proportions of overweight and obesity in this group (boys: 50.44%, girls: 36.73%) were also higher than those of underweight or normal weight at the 7th time period. Collectively, children experiencing AR during the 1st, 2nd, and 3rd time periods showed higher proportions of underweight, but lower proportions of overweight and obesity in the corresponding time periods. That is, children experiencing AR before the age of 2 years (1st~3rd time periods) showed the continuously increasing patterns of being overweight and obese. These patterns were more obvious when the AR occurred at an earlier time. 

Interestingly, children experiencing BMI rebound during the 4th, 5th, and 6th time periods showed a small increase in the proportion of underweight in the 2nd time period, but a dramatically higher proportion of underweight during the corresponding time period at AR onset, similar to an M-shaped patterns (Figure 2). On the other hand, the overweight or obese children in the above group showed a small decrease in the proportion of underweight in the 2nd time period, but a dramatically lower proportion of overweight and obesity during the corresponding time period of AR occurrence, similar to a W-shaped pattern. That is, children with BMI rebound after the age of 2 years experienced two episodes of being underweight and recovered, and then showed continuous increases in BMI. These patterns were more obvious when the AR occurred earlier. 

Among the children who experienced non-rebound of BMI up to the 7th time period, the proportion of underweight was slightly increased during the 2nd time period, but continuously increased during the seven time periods (Figure 2). On the other hand, the proportion of overweight and obesity was slightly decreased during the 2nd time period, but was gradually decreased during the seven time periods. That is, this group had a higher proportion of normal weight during the seven time periods than the other groups, which may indicate that they have not yet met the period of 2nd rebound of BMI.

### 3.4. Determinants of Early Adiposity Rebound 

Table 2 presents the nutrition-related parameters that may predict the risk of EAR at each period after adjusting for birth weight, household income level, and preterm birth. Using the generalized linear model, the input variables from the 1st to the 5th time period were checked to determine whether they significantly predicted the risk of EARs in the corresponding period.

In the 1st time period, there were no significant parameters predicting the risk of EAR among the input variables for boys. On the other hand, formula milk increased 23% of the EAR risk than feeding on breast milk alone (*p* = 0.009), and feeding on mixed milk also tended to increase the risk of EAR by about 14% among the girls (*p* = 0.076). In addition, start of weaning food after the age of 4 months tended to reduce the risk of EAR by 19% than that before the 4 months (*p* = 0.075).

In the 2nd time period, a 1 unit increase in the number of feeding baby food daily tended to increase the risk of EAR by 15% among the boys (*p* = 0.052). However, the risk of EAR was increased in feeding on soy milk by 47% (*p* = 0.021), and in feeding on cow’s milk by 55% (*p* = 0.014) than feeding on breast milk alone. Among the girls, the risk of EAR was also increased in feeding on soy milk by 46% (*p* = 0.016), and in feeding on cow’s milk by 51% (*p* = 0.009) than feeding on breast milk alone. 

In the 3rd time period, boys showed that the risk of EAR was significantly increased by a 1 unit increase in drinking sugar-sweetened beverages daily (15%, *p* = 0.001), by the use of salt in food (9%, *p* = 0.015), and by vitamin supplements (10%, *p* < 0.001). Among the girls, the risk of EAR was also significantly increased by a 1 unit increase in drinking sugar-sweetened beverages daily (8%, *p* = 0.002), by the use of salt in food (11%, *p* = 0.001), and by vitamin supplements (11%, *p* < 0.001). 

In the 4th time period, the risk of EAR was decreased by a 1 unit increase in appetite (9%, *p* < 0.001) and by a 1 unit increase in drinking cow’s milk (9%, *p* < 0.001) in boys. On the other hand, the risk of EAR was increased by a 1 unit increase in drinking sugar-sweetened beverages (7%, *p* = 0.075) and by parents’ concern about their child’s eating habits (7%, *p* = 0.009). Among the girls, the risk of EAR was decreased by the increased appetite (12%, *p* < 0.001), by increased snacking (4%, *p* = 0.043), and by higher consumption of cow’s milk (12%, *p* < 0.001), but decreased by drinking sugar-sweetened beverages (14%, *p* = 0.002).

In the 5th time period, a 1 unit increase in drinking sugar-sweetened beverages increased the risk of EAR (14%, *p* = 0.001) among boys. In addition, the risk of EAR was decreased by a 1 unit increase in the number of meals per day (9%, *p* = 0.002) and by a 1 unit increase in drinking cow’s milk (5%, *p* = 0.009) among the girls. On the other hand, a 1 unit increase in drinking sugar-sweetened beverages increased the risk of EAR by 12% (*p* = 0.032).

## 4. Discussion

This study showed that the average BMI rebound indicating AR was most common in the 6th time period (54–60 months), more than 50% of children experienced EAR before the age of 5 years, and the proportion of EAR was a little bit higher in girls than in boys (56.34% and 52.40%, respectively). Based on a previous study showing that the average age of AR is 5.5 years [9,11,24], Korean children in this study seemed to have earlier AR. In a recent French study, the proportion of AR before the age of 5.5 years was approximately 30% [25]. However, in the present study, 73.69% of children (boys: 73.01%, girls: 74.41%) experienced AR before the age of 5.5 years, which may indicate that EAR among Korean children is much higher than among French children.

It has been reported that children experiencing EAR are more likely to be exposed to the risk of being overweight or obese during growth [3,4,5,6,7,8,9]. In our study, approximately 18.27% of the boys and 23.60% of the girls had a risk of being underweight before the age of 2 years and then recovered their body weight. Additionally, more than half of all children (boys: 54.74% and girls: 50.81%) experienced two times the risk of being underweight and recovered before and after the age of 1 year, indicating an M-shaped pattern. That is, the risk of being underweight and recovering is much greater in the EAR onset and the proportion of being overweight and obese after the age of 5 years increased. 

Majority of children experiencing EAR (52.40% of boys and of 56.34% girls from the 1st to the 5th time period) showed similar body weight to those of the whole study population immediately after birth. However, they became overweight or obese during their growth, thereby having a large gap in BMI levels from those of the whole population. In addition, more than 40% of the infants with severe underweight (5th percentile) at birth experienced AR in the 1st time period, and were exposed to the greater risk of being overweight or obese after the age of 5.5 years, compared with the other AR groups. Therefore, it is important that underweight infants need to be monitored about the risk of EAR during their early life period, and children who are rapidly growing up during the early life need to maintain a body weight suitable for their growth rate [4,15,16,17,18].

Based on our results, the parameters predicting EAR risk in each time period were diverse. In the 1st time period, feeding on formula milk compared with breast milk increased the risk of EAR among girls. In the 2nd time period, feeding on soy milk or cow’s milk rather than breast milk increased the risk of EAR in both boys and girls, and frequent feeding of baby food per day also tended to increase the risk in boys. In the 3rd time period, drinking sugar-sweetened beverages, using salt in the food and vitamin supplementation increased the risk of EAR in both boys and girls. In the 4th time period, low appetite, and low consumption of cow’s milk increased the risk of EAR among children, and drinking sugar-sweetened beverages increased the risk in girls. In the 5th time period, consumption of sugar-sweetened beverages increased the risk of EAR in both boys and girls, and low consumption of cow’s milk and reduced meal frequency also increased the risk among girls. That is, adequate breast feeding before the age of 1 year, and adequate diet intake and reduced sugar-sweetened beverage consumption after the age of 1 year seems to be important for the prevention and reduction of EAR risk [4]. 

In this study, more than half of the infants experienced a weak or rapid reduction in BMI before the age of 1 year. In this period, children may have higher physical activity which can lead to increased energy expenditure or nutritional imbalance if breast milk is inadequately replaced with other types of diet [26]. Therefore, it is necessary to maintain stable breastfeeding in this period. As the growth rate of height continues in this period [17], providing adequate meals for normal growth is needed to prevent the rapid reduction in BMI. As the growth rate of height is maintained after the age of 1 year, and the rapid reduction in weight increment, particularly before the age of 2 years, contributes to the increased risk of overweight and obesity, an adequate supply of balanced nutrition during infancy is important [4]. Additionally, drinking sugar-sweetened beverages should be avoided because it can reduce the amount of foods that the infants have to consume [4].

Our results demonstrated that Korean children experienced AR in average before the age of 5 years, which is a little earlier than that in the Western children [9,10]. Additionally, the proportion of obese children is higher among boys and overweight children is slightly higher among girls based on the WHO standard. Previous studies have suggested that obesity [16], high calorie intake [23], and a Westernized diet [14] are predictive factors for EAR. However, obesity and high calorie intake may not be the cause of EAR, but rather its results. In addition, a westernized diet may contribute to the rapid increase in body weight after EAR, rather than being the cause itself [4]. Meanwhile, the majority of the non-AR group in the 7th time period is supposed to experience AR after this period, but it is necessary to follow up the pattern of their BMI up to adulthood.

Previous studies and our results showed that feeding on breast milk in early life may be a major predictor for lowering the risk of EAR [10,23,25]. Compared with the composition in the formula milk or cow’s milk, breast milk contains less protein and more fat but has higher proportion of unsaturated fatty acids [26,27,28,29]. Rolland-Cachera et al. [4,24] demonstrated that high protein and low-fat intake during early life could cause EAR. This may partly support our results showing the beneficial effect of breastfeeding on reducing the risk of EAR. However, we found that insufficient intake of cow’s milk after the age of 30 months may not help the growth and increase the risk of EAR. In addition, insufficient consumption of diet and snack including fruits, and higher intake of sugar-sweetened beverages may contribute to the increase of the risk of EAR. Therefore, it may be important to maintain normal body weight by supplying adequate amount of balanced nutrition during periods of rapid growth.

The limitations of this study are as follows. First, when the BMI percentiles of the WHO standards were applied to our study population, the interval between measurements was approximately 3 to 6 months. Thus, age was inevitably categorized based on the middle or middle plus a month during the period of measurement. Second, the children’s physical activity was not controlled, and data for BMI at birth was not presented. Third, this study focused on infancy, but need to track the BMI trajectory during the whole life cycle including fetal life particularly in the first trimester of pregnancy which can influence the later life [30,31]. In addition, this study applied to the WHO standard instead of the Korean National Growth Chart for children and adolescents (2017 updated), because this study used data of the population born in 2008, 2009, and 2010 which was close to the period that the WHO standard was reported. In the future study, we need to apply to the Korean standard.

## 5. Conclusions

Despite these limitations, this study is meaningful for presenting the onset of AR and the major determinants of EAR among Korean infants and children for the first time based on large-scale nationwide data. These results can also provide a basis for establishing management guidelines to prevent children from becoming obese and related chronic degenerative diseases in the future.

## Figures and Tables

**Figure 1 nutrients-14-00929-f001:**
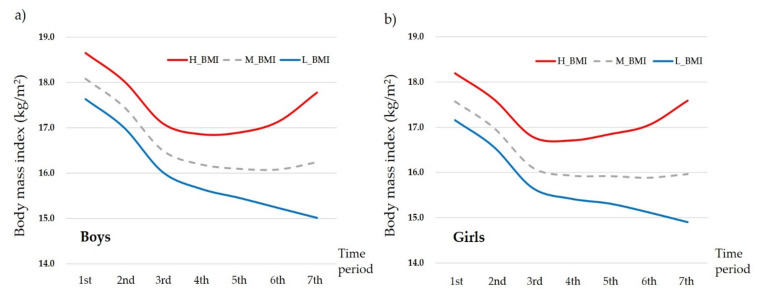
Changes in average body mass index (BMI) for the seven time periods by (**a**) boys and (**b**) girls. H_BMI: higher BMI group whose BMI at the 7th time period was higher than the average BMI at the 7th time period; L_BMI: lower BMI group whose BMI at the 7th time period was lower than the average BMI at the 7th time period; M: average BMI group based on the 7th time period.

**Figure 2 nutrients-14-00929-f002:**
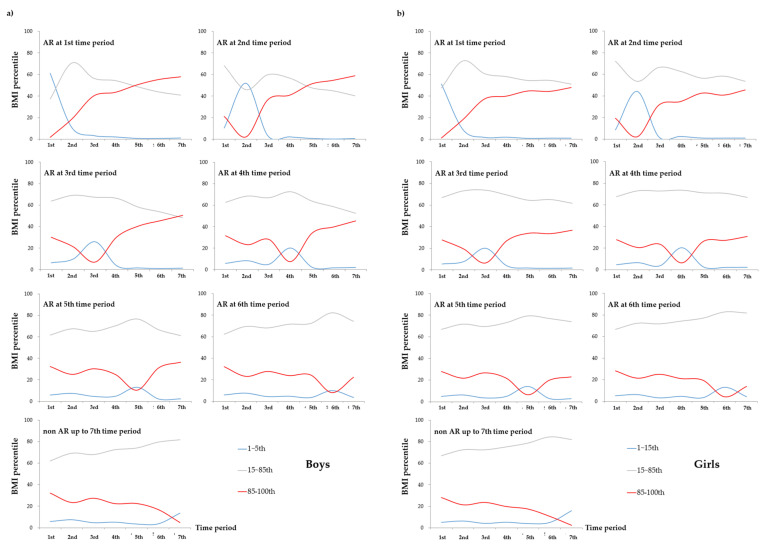
Distribution of body mass index (BMI) percentile for the seven-time periods according to each time period and onset of adiposity rebound (AR) by (**a**) boys and (**b**) girls. The determination of the BMI percentile by age and sex was based on the child growth standards of the World Health Organization. Specifically, BMI-for-age less than 5th percentile is categorized as severely underweight, 5th to less than 15th percentile into underweight, 15th to less than 85th percentile into healthy weight, 85th to less than 95th percentile into overweight, and 95th to 100th percentile into obese based on the age at the time of measurement.

**Table 1 nutrients-14-00929-t001:** Basic information of study population.

Age at Survey (Time Period)	Parameters	Unit/Category	Total (*n* = 142,668)	Boys (*n* = 73,389)	Girls (*n* = 69,279)
4~6 month (1st)	Birth weight	Kg	3.20 ± 0.46	3.25 ± 0.47	3.15 ± 0.45
	Household income	1~20 (5–95%)	11.81 ± 4.44	11.81 ± 4.46	11.82 ± 4.42
	Preterm birth	Yes	3.67	3.61	3.72
	Type of feeding milk	breast milk	46.32	44.33	48.42
		formula milk	32.94	33.82	32.01
		mixed milk	20.35	21.42	19.21
		special formula milk	0.40	0.43	0.36
	Start of weaning food	before 4 months	6.70	6.85	6.55
		after 4 months	93.30	93.15	93.45
9~12 month (2nd)	Number of weaning food per day	1 time	2.51	2.43	2.59
		2 times	25.29	24.27	26.37
		3 times	69.58	70.62	68.46
		4 or more times	2.62	2.67	2.57
	Type of feeding milk	breast milk	39.76	38.81	40.77
		goat’s milk	2.91	2.99	2.83
		soy milk	3.96	4.05	3.86
		cow’s milk	4.34	4.25	4.43
		formula milk	49.02	49.88	48.10
18~24 month (3rd)	Daily intake of sugar-sweetened beverages	200 mL or less	93.79	93.40	94.20
		200–500 mL	5.76	6.12	5.37
		500 mL or more	0.46	0.48	0.43
	Use of salt in food	yes	83.61	83.90	83.30
	Vitamin supplements	yes	38.63	39.36	37.85
30~36 month (4th)	Appetite	bad	5.40	5.55	5.24
		normal	50.73	50.68	50.79
		good	43.87	43.77	43.97
	Number of meals per day	1 time	0.33	0.35	0.31
		2 times	9.83	9.18	10.51
		3 times	88.53	89.14	87.89
		4 or more times	1.31	1.33	1.29
	Number of snacks per day	1 time	7.99	7.96	8.03
		2 times	67.65	67.82	67.48
		3 or more times	24.35	24.22	24.49
	Daily intake of cow’s milk	200 mL or less	33.78	32.42	35.21
		200–500 mL	60.25	61.10	59.34
		500 mL or more	5.97	6.47	5.45
	Daily intake of sugar-sweetened beverages	200 mL or less	93.84	93.20	94.53
		200–500 mL	5.50	6.10	4.86
		500 mL or more	0.66	0.69	0.62
	Parents’ concern about child’s eating habits	Yes	47.69	48.15	47.20
42~48 month (5th)	Number of meals per day	1 time	0.33	0.38	0.38
		2 times	9.83	6.88	8.38
		3 times	88.53	91.34	90.04
		4 or more times	1.31	1.39	1.20
	Daily intake of cow’s milk	200 mL or less	37.66	35.39	40.07
		200–500 mL	58.72	60.44	56.90
		500 mL or more	3.61	4.16	3.03
	Daily intake of sugar-sweetened beverages	200 mL or less	94.08	93.32	94.89
		200–500mL	5.67	6.40	4.90
		500 mL or more	0.25	0.28	0.22

Results were presented as means ± standard deviation or %.

**Table 2 nutrients-14-00929-t002:** Determinants of early adiposity rebounds among boys and girls.

**Boys (*n* = 73,389)**					
Parameters	Age at Survey (Time Period)	B ± SE	β	OR	*p*
Number of weaning food per day	9~12 month (2nd)	0.14 ± 0.07	0.33	1.15	0.052
Soy milk (referring to breast milk)		0.38 ± 0.17	0.33	1.47	0.021
Cow’s milk (referring to breast milk)		0.44 ± 0.18	0.34	1.51	0.009
Daily intake of sugar-sweetened beverages	18~24 month (3rd)	0.14 ± 0.04	0.09	1.15	0.001
Use of salt in food		0.08 ± 0.03	0.07	1.09	0.015
Vitamin supplements		0.09 ± 0.02	0.11	1.10	<0.001
Appetite	30~36 month (4th)	−0.09 ± 0.02	−0.12	0.91	<0.001
Daily intake of cow’s milk		−0.09 ± 0.02	−0.12	0.91	<0.001
daily intake of sugar-sweetened beverages		0.07 ± 0.04	0.04	1.07	0.075
Parents’ concern about child’s eating habits		0.06 ± 0.02	0.07	1.07	0.009
Daily intake of sugar-sweetened beverages	42~48 month (5th)	0.13 ± 0.04	0.08	1.14	0.001
**Girls (*n* = 69,279)**					
**Parameters**	**Age at Survey (Time Period)**	**B ± SE**	**β**	**OR**	** *p* **
Formula milk (referring to breast milk)	4~6 month (1st)	0.22 ± 0.08	0.37	1.23	0.009
Mixed milk (referring to breast milk)		0.13 ± 0.07	0.26	1.14	0.076
Start of weaning food after 4 months		−0.21 ± 0.12	−0.22	0.81	0.075
Soy milk (referring to breast milk)	9~12 month (2nd)	0.38 ± 0.16	0.27	1.46	0.016
Cow’s milk (referring to breast milk)		0.41 ± 0.16	0.28	1.51	0.009
Daily intake of sugar-sweetened beverages	18~24 month (3rd)	0.13 ± 0.04	0.08	1.08	0.002
Use of salt in food		0.10 ± 0.03	0.09	1.11	0.001
Vitamin supplements		0.11 ± 0.02	0.12	1.11	<0.001
Appetite	30~36 month (4th)	−0.12 ± 0.02	−0.15	0.88	<0.001
Number of snacks per day		−0.04 ± 0.02	−0.05	0.96	0.043
Daily intake of cow’s milk		−0.12 ± 0.02	−0.15	0.88	<0.001
Daily intake of sugar-sweetened beverages		0.14 ± 0.04	0.08	1.14	0.002
Number of meals per day	42~48 month (5th)	−0.10 ± 0.04	−0.07	0.91	0.002
Daily intake of cow’s milk		−0.06 ± 0.02	−0.07	0.95	0.009
Daily intake of sugar-sweetened beverages		0.11 ± 0.05	0.06	1.12	0.032

B ± SE: beta coefficient ± standard error; β: adjusted beta coefficient; OR: odds ratio; *p*: *p*-value.

## Data Availability

The data presented in this study are publicly available, but after the approval from the institutional review board.

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
