# Peer review of "Timing of Adiposity Rebound and Determinants of Early Adiposity Rebound in Korean Infants and Children Based on Data from the National Health Insurance Service"

_nutrients, 2022, doi:10.3390/nu14050929_

Round 1

Reviewer 1 Report

The aim of the paper is to identify nutritional and dietary factors related to early adiposity rebound (AR) on a large cohort of Korean children. An important topic in the prevention of overweight/obesity.

Some comments:

  1. The authors state that the children in the study population were classified into "7 AR groups according to the specific time period when the BMI level became minimum  and then increased again" . This indicates that they had to have at least two anthropometric measurements within each specific time period in order to observe first the minimum BMI level and then again observe the increase during the same period. This leads to question: how many measurements were done within each time period? The figures were obviously based on smoothing splines of these measurements which look good.
  2. What about if the minimum point of BMI was taking place in one time period but the increase was observed only in a later period, when do you then consider the child having an AR?
  3. How did you assure that the weight/height measurements across the hospitals were comparable?
  4. How did the low-birth weight babies experience the AR? Did you look at it? Now you only adjust for low birth weight.
  5. The WHO BMI standards were used but in Fig. 1 the BMI values were categorized as high, and low but the cut off points were not given.
  6. In Fig. 2 the Y-axis describes the percentile of the BMI, not BMI. 
  7. Are WHO BMI standards appropriate for a Korean population? Any Korean specific standards available? WHO standards have received some criticism that they are  applicable to Western populations only. They may over/underestimate malnutrition, either undernutrition or overnutrition. Could you add discussion regarding it. 
  8. The main aim of the study was to identify nutritional factors associated with AR. However, the nutritional factors are not described well. E.g. what are 'sugary beverages' in the Korean food culture? Did the parents know what to include into this food group? What do you mean with fresh milk (unprocessed cow's milk from grocery store?)? Why only vitamin supplements? For example use of iron supplements are fairly common in children. Were these dietary supplements over-the-counter or prescribed by a physician? What is the difference between formula milk and special formula milk?
  9. There are some typos still in the text. 
  10. What does the '*' refer to in Table3?

Author Response

Answers for Reviewer’s comments

Manuscript Number: Nutrients-1597517.R1

Timing of adiposity rebound (AR), and determinants of early AR in Korean infants and children based on the data from the National Health Insurance Service

Dear Reviewer #1

We sincerely appreciate the time spent in reviewing this manuscript and your advice to improve it.

Please, see below our answers to your queries and comments. We also marked the corrected and revised parts of the text with red. We hope that you find them satisfactory.

Sincerely yours,

Hyo Jeong Jeon

Comments from Reviewer #1:

The aim of the paper is to identify nutritional and dietary factors related to early adiposity rebound (AR) on a large cohort of Korean children. An important topic in the prevention of overweight/obesity.

Some comments:

1) The authors state that the children in the study population were classified into "7 AR groups according to the specific time period when the BMI level became minimum and then increased again". This indicates that they had to have at least two anthropometric measurements within each specific time period in order to observe first the minimum BMI level and then again observe the increase during the same period. This leads to question: how many measurements were done within each time period? The figures were obviously based on smoothing splines of these measurements which look good.

Answer) The authors sincerely appreciate the reviewer’s comment for improving this paper. Height and weight of each child were measured once during each time period by a trained medical staff in the hospital. As mentioned in the discussion, the interval between measurements was approximately 3 to 6 months. Thus, age was inevitably categorized based on the middle or middle plus a month during the period of measurement. Smoothing splines in the figures were made by the option. When we draw the line without smoothing potion, we found the similar pattern in the figure.

2) What about if the minimum point of BMI was taking place in one time period but the increase was observed only in a later period, when do you then consider the child having an AR?

Answer) According to the previous reports, the average age of AR was 5.5 - 6.6 years (5.5 years in the US cohort data, and 6.6 years for boys and 6.0 years for girls in the New Zealand cohort data). In our current study, the average AR was found before the age of 5 years (54-60 months). Based on these results, we assumed that non-AR group up to 7th time period (66-71 months) may be in the period of AR or have AR at age of 6 years, right after the 7th time period (72months~). Unfortunately, we cannot collect the data after the 7th time period. So, it is needed to do further study for tracking the BMI trajectory of the children at least by their age up to 20 years.

3) How did you assure that the weight/height measurements across the hospitals were comparable?

Answer) As mentioned in the above, the trained medical staffs in each hospital measured height and weight of a child following the guideline. The bias in the measurement across the hospitals may exist, but could be neglectable by the large-scale data.

4) How did the low-birth weight babies experience the AR? Did you look at it? Now you only adjust for low birth weight.

Answer) The authors thank you for your comments. We examined the BMI trajectory in the low-birth weight babies (< 2.5kg at birth, 6.2% of total study population) over the seven time periods. As shown in the Table below, we found that the proportion of AR among the low-birth weight babies at each time period is similar with that normal-birth weight babies (91.42%) and large-birth weight babies (³4.5kg at birth, 2.35% of total study population). Proportion of adiposity rebound occurrence at each time period (1st to 7th) was 2.8 %, 3.3 %, 13.3 %, 18.5 %, 16.5 %, 20.0 % and 26.6 % in normal birth weight babies, 2.5 %, 3.8 %, 14.6 %, 17.4 %, 15.9 %, 19.4 %, and 26.4 % in the low-birth weight babies, and 3.2 %, 4.8 %, 16.5 %, 16.3 %, 17.2 %, 17.9 %, and 24.1 % in the large-birth weight babies, respectively. Nevertheless, we adjusted for low birth weight to increase the statistical power.

body weight at birth

Proportion (%) of adiposity rebound occurrence at each time period

1st

2nd

3rd

4th

5th

6th

7th

Sub total

normal

2.8

3.3

13.0

18.5

16.5

20.0

26.0

100.0

low

2.5

3.8

14.6

17.4

15.9

19.4

26.4

100.0

large

3.2

4.8

16.5

16.3

17.2

17.9

24.1

100.0

5) The WHO BMI standards were used but in Fig. 1 the BMI values were categorized as high, and low but the cut off points were not given.

Answer) The author are sorry for making the reviewer confused. We clearly explained the meaning of the words in the footnote of Figure 1.

 “H_BMI: higher BMI group whose BMI at the 7th time period was higher than the average BMI at the 7th time period; L_BMI: lower BMI group whose BMI at the 7th time period was lower than the average BMI at the 7th time period; M: average BMI group based on the 7th time period.’

6) In Fig. 2 the Y-axis describes the percentile of the BMI, not BMI. 

Answer) The authors are sorry for making the reviewer confused. We changed it to “BMI percentile”.

7) Are WHO BMI standards appropriate for a Korean population? Any Korean specific standards available? WHO standards have received some criticism that they are applicable to Western populations only. They may over/underestimate malnutrition, either undernutrition or overnutrition. Could you add discussion regarding it. 

     Answer) The authors deeply appreciate your comment. As you commented, we have the Korean National Growth Chart for children and adolescents (updated in 2017). However, we used data of the population born in 2008, 2009, and 2010 which had been tracked from the 1st time period (4-6 months) to the 7th time period (66-71 months). Therefore, we thought that it was reasonable to use the WHO BMI standard which was close to the period of examination. In the discussion section, we mention the necessity of applying to the Korean standard in the future study.

8) The main aim of the study was to identify nutritional factors associated with AR. However, the nutritional factors are not described well. E.g. what are 'sugary beverages' in the Korean food culture? Did the parents know what to include into this food group? What do you mean with fresh milk (unprocessed cow's milk from grocery store?)? Why only vitamin supplements? For example use of iron supplements are fairly common in children. Were these dietary supplements over-the-counter or prescribed by a physician? What is the difference between formula milk and special formula milk?

     Answer) The authors thank you again on your comment for improving this manuscript. the authors briefly explained the input variables related to nutrition in some questionnaires for each time period in the method section. As mentioned in the manuscript, the questionnaires were self-reported by the parents of the child. Shortly, the words ‘sugary beverage’, and ‘fresh milk’ were revised to ‘sugar-sweetened beverages’, and ‘cow’s milk’, respectively. In addition, the questionnaire for supplement consumption included only vitamin supplements without detailed questions. Special formula milk indicates the formulas typically made to help with a specific health or feeding issue.

9) There are some typos still in the text. 

    Answer) The authors are sorry for making the reviewer confused. We corrected all the typos in the text.

10) What does the '*' refer to in Table3? 

Answer) * indicates that snacks includes fruits. To avoid reader’s confusion, the authors deleted * in the table, and explained it in the method section. “…number of snacks (including fruits) per day…”

Reviewer 2 Report

In this study, the authors aimed to investigate patterns of childhood AR during 7 time periods and the nutritional determinants of early AR in a large cohort of Korean infants. This is an interesting study that has high relevance in the area of childhood obesity and predictive patterns of obesity later in life. The strengths include using a large cohort for the study over a considerable time period. The data is presented well in a concise but detailed way that is easy to read.

My main concern throughout is the weak level of English language. In many places I was unsure whether I understood what the authors were trying to convey, particularly in the results section. The discussion is well written however lacks reference to the most relevant literature in the field and suggestions for future work.

Some minor comments:

Briefly define the term "adiposity rebound" in the abstract.

The title could be written better to avoid the use of abbreviations. Remove the word the before data.

Line 59: this is an exaggerated statement. It is better to say "associated with" than 'caused'.

In line 120: give some details of the questionnaire. Perhaps this could be added to the supplementary file.

Overall, I commend the authors on a well-designed study that is presented well. To improve, I advise the authors to revise the English language throughout.

Author Response

Answers for Reviewer’s comments

Manuscript Number: Nutrients-1597517.R1

Timing of adiposity rebound (AR), and determinants of early AR in Korean infants and children based on the data from the National Health Insurance Service

Dear Reviewer #2

We sincerely appreciate the time spent in reviewing this manuscript and your advice to improve it.

Please, see below our answers to your queries and comments. We also marked the corrected and revised parts of the text with red. We hope that you find them satisfactory.

Sincerely yours,

Hyo Jeong Jeon

Comments from Reviewer #2:
In this study, the authors aimed to investigate patterns of childhood AR during 7 time periods and the nutritional determinants of early AR in a large cohort of Korean infants. This is an interesting study that has high relevance in the area of childhood obesity and predictive patterns of obesity later in life. The strengths include using a large cohort for the study over a considerable time period. The data is presented well in a concise but detailed way that is easy to read. My main concern throughout is the weak level of English language. In many places I was unsure whether I understood what the authors were trying to convey, particularly in the results section. The discussion is well written however lacks reference to the most relevant literature in the field and suggestions for future work.

Answer) The authors sincerely appreciate the reviewer’s comments. In accordance with your advice, we revised the discussion part by adding relevant literatures, and also revised English language throughout the manuscript by an English expert (EDTAGE Job code: RMSUH_3).

Some minor comments:

1) Briefly define the term "adiposity rebound" in the abstract.

Answer) In accordance with the reviewer’s advice, the authors briefly defined the term ‘adiposity rebound’ in the abstract.

2) The title could be written better to avoid the use of abbreviations. Remove the word the before data.

Answer) As you commented, we revised the title.

3) Line 59: this is an exaggerated statement. It is better to say "associated with" than 'caused'.

Answer) As you commented, we revised it.

4) In line 120: give some details of the questionnaire. Perhaps this could be added to the supplementary file.

Answer) As the questionnaires for each time period includes plenty numbers of items (1st time period: 63 variables, 2nd time period: 81, 3rd time period: 80, 4th time period: 91, 5th time period: 99, 6th time period: 101, and 7the time period: 98), we could not add all of them in the supplementary file. Instead, the authors briefly explained the input variables related to nutrition in some questionnaires for each time period in the method section. 

“Briefly, the input variables related to nutrition in the questionnaires for each time period were as follows: type of feeding milk (i.e. breast milk, formula milk, mixed milk including breast milk and formula milk, and special formula milk) and start of weaning food before or after the age of 4 months during the 1st time period; number of weaning food per day, timing of the 2nd start of weaning food and type of milk (i.e. breast milk, goat’s milk, soy milk, cow’s milk and formula milk), and any of ingredients (i.e. grains, vegetables, eggs, fish, meats) omitted in weaning food at the 2nd time period; regular eating at the given place, daily intake of sugar-sweetened beverages, use of salt in food, and vitamin supplementation at the 3rd time period; appetite, number of meals per day, number of snacks (including fruits) per day, daily intake of cow’s milk, daily intake of sugar-sweetened beverage, parents’ concerning about child’s eating habits at the 4th time period; appetite, number of meals per day, daily intake of cow’s milk, daily intake of sugar-sweetened beverage, and picky eating at the 5th time period.”

5) Overall, I commend the authors on a well-designed study that is presented well. To improve, I advise the authors to revise the English language throughout.

Answer) In accordance with your advice, we revised the English language throughout the manuscript by an English expert (EDTAGE Job code: RSUH_3).